# Investigation into the Direct Photolysis Process of Photo-Induced RAFT Polymerization by ESR Spin Trapping

**DOI:** 10.3390/polym11101722

**Published:** 2019-10-21

**Authors:** Jiajia Li, Mengmeng Zhang, Jian Zhu, Xiulin Zhu

**Affiliations:** 1State and Local Joint Engineering Laboratory for Novel Functional Polymeric Materials, Jiangsu Key Laboratory of Advanced Functional Polymer Design and Application, Department of Polymer Science and Engineering, College of Chemistry, Chemical Engineering and Materials Science, Soochow University, Suzhou 215123, China; 20154209023@stu.suda.edu.cn (J.L.); 20184209174@stu.suda.edu.cn (M.Z.); xlzhu@suda.edu.cn (X.Z.); 2Global Institute of Software Technology, Suzhou 215163, China

**Keywords:** visible light, spin-trapping, ESR, RAFT

## Abstract

The direct photolysis of reversible addition fragmentation chain transfer (RAFT) agents under visible light was demonstrated by electron spin resonance (ESR) using 5,5-dimethyl-1-pyrroline *N*-oxide as a typical spin trap. The hyperfine coupling lines obtained by ESR spectroscopy showed the successful capture of the carbon-centered and the sulfur-centered radical. Photo-polymerization of vinyl acetate under different wavelengths was performed to verify the effects of wavelength on the process. The effect of the R group of RAFT agents on the photolysis was investigated by spin-trapping experiments using poly (butyl acrylate) and poly (vinyl acetate) as macroRAFT agents. The quantitative experiment showed the yield of photolysis of a xanthate to be only 0.023% under λ > 440 nm.

## 1. Introduction

Visible light has been widely applied in various reversible deactivation radical polymerization (RDRP) [1] techniques, such as nitroxide mediated polymerization (NMP) [2,3,4,5], atom-transfer radical polymerization (ATRP) [6,7,8,9] and reversible addition fragmentation chain transfer (RAFT) [10,11,12] polymerization, owing to its advantages of mild reaction conditions, cheap commercially available light sources, and particularly, the spatial and temporal control [13] of the polymerization. Among these methods, visible-light-induced RAFT [14] polymerization has been considered as one of the most versatile RDRP techniques due to its tolerance of a wide range of compatible reagents.

Currently, three main approaches are known for photo-RAFT polymerizations. The first approach utilizes photo initiators instead of traditionally used thermal initiators for generating primary radicals. For instance, Cai′s group developed photo-RAFT polymerization carried out under UV light and sunlight by using diphenyl(2,4,6-trimethylbenzoyl)phosphine oxide (TPO) and its derivatives as photo initiators [15,16]. The second approach is the activation of a RAFT agent using a photoredox catalyst, namely photo-induced electron transfer RAFT (PET-RAFT) polymerization [17], developed by Boyer and co-workers. In this approach, the primary radical is generated by the reduction of a RAFT agent by a photoredox catalyst under visible light irradiation. Various photoredox catalysts have been explored for this process, including Ir(ppy)_3_ [18], Ru(bpy)_3_Cl_2_ [19], ZnTPP [20], and organo dyes [21]. The main advantages of the use of such catalysts are their oxygen tolerance [22] and wavelength selectivity [23]. The former allows for the controlled polymerizations to be carried out without degassing, thus making the process more convenient. The dissolved oxygen can be consumed by catalysts owing to its strong photo-reducing nature. The wavelength selectivity has been applied in single unit monomer insertion (SUMI) reactions to prepare sequence-defined polymers by selecting appropriate monomer/chain transfer agent and catalyst/wavelength combinations [24]. Finally, the third approach of photo-RAFT involves the direct photoactivation of RAFT agents without an exogenous photocatalyst. The advantages of this method include high chain end fidelity and ease of operation due to simple components (monomer and RAFT agent). Furthermore, no toxic metal- or organo-photocatalysts are needed for the reactions.

The direct photoactivation of thio compounds is well known and was first reported by Otsu as photoiniferters [25], paving the way for their subsequent use in polymerization. Soon after the first report on RAFT polymerization [10], the activation of RAFT agents via irradiation was investigated. Bai et al. reported the first RAFT polymerization under γ-radiation [26], which can produce propagating radicals from a monomer, RAFT agent, or solvent. With the development of the concept of “green chemistry”, UV irradiation-induced photo-RAFT polymerization later emerged as a milder alternative to γ-radiation [27,28,29]. Since then, the activation of RAFT agents under visible light has gained much attention. Recent studies have utilized the n→π* transition absorption of the C-S double bond of RAFT agents using 400–600 nm light [30]. In 2015, the Boyer′s group and Qiao′s group both reported such photo-RAFT through the direct photoactivation of RAFT agents [31,32], followed by several similar reports, investigating various monomers and RAFT agents [33,34]. In the reported systems, the reversible termination process provided the initial propagating radicals and was also considered a reversible degenerative process, playing a similar role to the degenerative transfer process. The mechanism of the direct photoactivation process was proposed as a combination of reversible termination and degenerative transfer (Scheme 1). Subsequently, the possibility of oxygen tolerance of this method was researched [35,36] and was later referred to as “polymerizing through” oxygen [37]. The dissolved oxygen was hypothesized to be consumed by the RAFT agent, as suggested by a longer induction time. In the meantime, the wavelength selectivity has also been developed for the synthesis of complex polymeric materials through careful selection of the wavelength of the irradiation and radical stabilization energy (RSE) of the R-leaving group [38]. However, the mechanism of this direct photoactivation process is still unclear due to limited research that has mainly focused on the photolytic stability of RAFT agents [39] and the effects of light sources on the apparent polymerization rate [40]. There is still no direct evidence to prove the photolysis of the C–S bond in RAFT agents. Especially, the reversible termination and degenerative transfer process need to be further understood.

Electron spin resonance (ESR) is a selective and direct method for detecting free radicals [41]. However, using this approach, the direct detection of radicals generated by the photo-RAFT process is difficult due to the low radical concentration. The ESR spin-trapping technique [42] has, however, been extensively used in such cases. 5,5-Dimethyl-1-pyrroline *N*-oxide (DMPO) is typically used as a spin trap by radical addition to form more stable radical adducts, which can be detected by ESR spectroscopy [43,44]. The signals in ESR spectra from DMPO adducts can be used to determine the structures of radical adducts based mainly on the hyperfine coupling constants, which are, in general, useful for distinguishing trapped radicals. Herein, we employ DMPO as a spin trap for the radicals generated from the direct photolysis of a RAFT agent under visible light, aiming to prove the direct photoactivation process (Scheme 2).

## 2. Materials and Methods 

### 2.1. Materials

Vinyl acetate (VAc) and n-butyl acrylate (BA) were obtained from Chinasun Specialty Products Co., Ltd. (Shanghai), China at AR grade. They were purified by passing through a neutral alumina column before polymerization. (2,2,6,6-Tetramethylpiperidin-1-yl)oxyl (TEMPO), 5,5-Dimethyl-1-pyrroline *N*-oxide (DMPO), 2-cyanopropan-2-yl dodecyl carbonotrithioate (CPDT), and 2-cyanopropan-2-yl benzodithioate (CPDB)) were ordered from Sigma–Aldrich (Shanghai, China). These chemicals were used without further purification. Ethyl 2-((ethoxycarbonothioyl)thio)propanoate (EXEP) was synthesized according to the literature [45]. 

### 2.2. Polymerization Procedures

The polymerization of VAc is described as a typical example. The monomer (1 mL, 10.8 mmol) was mixed with EXEP (12 mg, 0.054 mmol).The mixture was placed in an ampoule bottle. The content was degassed by three freeze–pump–thaw cycles under nitrogen atmosphere. The polymerization was carried out by blue or purple LED irradiation at a controlled temperature. After the desired polymerization time, the ampoule was exposed to oxygen. The product was diluted by tetrahydrofuran, and the polymer was obtained after precipitation in hexane. Monomer conversion was calculated by gravimetric analysis.

### 2.3. Characterization

TOSOH HLC-8320 size exclusion chromatography (SEC) was used to determine the number-average molecular weight (*M*_n_) and molecular weight dispersity (*Ð*) of polymers equipped with a TSK gel Muliti pore HZ-N (3) 4.6 mm × 150 mm column at 40 °C. THF was used as the eluent with a flow rate of 0.35 mL min^−1^. SEC samples were injected using a TOSOH HLC-8320 SEC plus auto sampler (Shanghai, China). Narrow polydispersity polystyrene standards were used to calibrate the molecular weights.

ESR measurements were carried out on a JEOL JES-X3 Series ESR spectrometer (Shanghai, China) at room temperature. General instrument parameters are as follows: microwave power, 1 mW, macrowave frequency, 9147 MHz, modulation frequency, 100 kHz, modulation amplitude, 300 G, field scan range, 100 G, center field, 3260 G. The photoirradiation is from ES-USH500 UV lamp (Shanghai, China) with power of 1000 W. The hyperfine coupling constants of the spin adducts were determined by simulating the spectra using the WinSim software (National Institute of Environmental Health Sciences, 0.96, Durham, NC, USA). 

## 3. Results and Discussion

RAFT agents with different structures, namely ethyl 2-((ethoxycarbonothioyl)thio)propanoate (EXEP), 2-cyanopropan-2-yl dodecyl carbonotrithioate (CPDT) and 2-cyanopropan-2-yl benzodithioate (CPDB) were employed for the spin trapping experiments after their UV-Vis spectra were obtained. The n→π* transition absorption of these RAFT agents can be seen in Figure 1. The Z group of a RAFT agent has an important effect on the absorption, as suggested by CPDB, with a resonance Z group showing a red shift of n→π* absorption, as opposed to those of CPDT and EXEP.

A reaction mixture of the RAFT agent and three equivalents of DMPO was then prepared and placed in the cavity of an ESR spectrometer, which could be directly irradiated by a light source in the spectrometer. A filter was used to cut undesired wavelengths, and the experiment was first studied under light irradiation of light with λ > 440 nm. Two series of six hyperfine lines for all three RAFT agents (Figure 2A) were observed, resulting from hyperfine coupling to one proton and one nitrogen. The result of the spin trapping of DMPO and EXEP was consistent with the simulated spectra (Figure 2B). Signal “a” was assigned to the adduct of DMPO and the carbon-centered radical with coupling constants of *a*_N_ = 13.90 G and *a*_β-H_ = 21.70 G while signal “b” was assigned to the adduct of DMPO and the sulfur-centered radical with coupling constants of *a*_N_ = 12.96 G and *a*_β-H_ = 13.85 G according to the literature [46,47]. Regarding the spin trapping result of CPDT, the intensity of signal “c” is lower than that of signal “d”, suggesting a lower reactivity between the isobutylcyanide radical and DMPO than the sulfur-centered radical. The direct photolysis of CPDB was also supported by the spin trapping by DMPO, despite the signals of “e” and “f” being very weak. Controlled experiments were performed by irradiating only DMPO or EXEP with λ > 440 nm light. No obvious signals were observed (Appendix A), which can be attributed to the recombination of radicals after photolysis without DMPO. It can be concluded that the signals detected in Figure 2 were indeed formed by successful spin-trapping of radicals generated from direct photolysis of RAFT agents under visible light irradiation.

EXEP was selected for further spin-trapping experiments due to its ESR signals being much clearer than those given by CPDT and CPDB. The effect of wavelength was investigated by irradiating the sample under λ > 390 nm, λ > 440 nm, and λ > 520 nm light (Figure 3A). The highest signal intensity of DMPO adduct was observed under light with λ > 390 nm, indicating that the photolysis of EXEP was most efficient under this wavelength. The relationship between signal intensity and time under different wavelengths is presented in Figure 3B. When λ > 390 nm light was used, the signal intensity rapidly reached a maximum and then started to diminish, suggesting the radical adduct was unstable under this wavelength. In comparison, upon irradiation with λ > 440 nm light, the signal intensity did not reach its maximum until after 3 min, followed by a slower decrease. With λ > 520 nm light, only a very weak signal was observed. Hence, it was reasonable to propose that more reversible termination took place under light with λ > 390 nm, which was hypothesized to have an accelerating effect on the polymerization rate.

Although the photo-polymerization of vinyl acetate (VAc) with EXEP was successfully performed under blue LED [33], the n→π* absorption of EXEP at around 360 nm suggested a purple LED (λ_em,max_ = 390 nm) was more appropriate for the polymerization. Kinetic experiments were performed to verify the effects of wavelength on the photo-polymerization. As expected, polymerization of VAc with EXEP under the irradiation of purple LED (λ_em,max_ = 390) showed a faster polymerization rate than that under the irradiation of blue LED (λ_em,max_ = 440) (Figure 4A). Molecular weights determined by SEC correlated well with theoretical molecular weights and increased linearly with monomer conversion (Figure 4B), indicating living polymerization under these wavelengths. The results were consistent with the spin-trapping experiments.

Next, BA or VAc was added to the mixture of EXEP and DMPO to investigate the photo-RAFT process in more depth. After irradiation by light with λ > 440 nm, the signal showed no obvious differences in the presence or in the absence of BA (Figure 3A and Figure 5A), while the signal of DMPO adduct generated from the sulfur-centered radical disappeared after 5 seconds under irradiation (Figure 5B) in the presence of VAc. We proposed that the structure of the R group changes from BA-type to VAc-type after radical addition of the monomer, thus affecting the photolytic stability of the RAFT agent.

PBA (*M*_n_ = 3500 g mol^−1^, *Ð* = 1.41) and PVAc (*M*_n_ = 1800 g mol^−1^, *Ð* = 1.06) prepared through photo-induced polymerization with EXEP were then used as macroRAFT agents for spin trapping experiments. Strong signal intensity of six hyperfine lines was observed from the PBA-DMPO adduct, in contrast to the weak signal intensity of the PVAc-DMPO adduct (Figure 6), further confirming that the structure of the R group has an important effect on the photolytic stability. 

Finally, a quantitative experiment was performed by the integral of the signal in the ESR spectrum, which makes it possible to calculate the quantum yield of the photolysis of the RAFT agent. A quantitative (2,2,6,6-Tetramethylpiperidin-1-yl)oxyl (TEMPO) was prepared and used as a standard sample. Hence, the concentration of DMPO-adduct can be obtained by comparing the integral of their signals via ESR software (Figure 7). The result showed the yield of photolysis of EXEP was only 0.023% under λ > 440 nm, implying the degenerative transfer process plays a key role in photo-induced RAFT polymerization.

## 4. Conclusions

The direct photolysis of RAFT agents under visible light was demonstrated using the spin-trapping technique with ESR for the first time. The quantitative experiment indicated that the degenerative transfer process plays the main role in the photo-RAFT process, while the reversible termination process was affected by the selectivity of the light wavelength. The structure of the R group of the RAFT agent had a significant effect on the photolytic stability. We demonstrated that the photo-RAFT polymerization was initiated by the carbon-centered radical, and reversibly terminated by the sulfur-centered radical. Thus, this work provided a further understanding of the direct photoactivation of RAFT agents under visible light.

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
