# Peer review of "Investigation into the Direct Photolysis Process of Photo-Induced RAFT Polymerization by ESR Spin Trapping"

_polymers, 2019, doi:10.3390/polym11101722_

Round 1

Reviewer 1 Report

In this manuscript the authors used the spin-trapping ESR technique to investigate the photo-induced polymerization process of vinyl acetate.

The manuscript is well organized and the experimental design is appropriate. However, in my opinion, it should be improved before publication since the analysis of the experimental data is not accurate and the corresponding conclusion are not completely correct. 

In particular, the authors should address these specific remarks:

line 16 (and others in the text): Please substitute “noxide”  with “N-oxide” lines 115-117: The experimental parameters are clearly wrong lines 129-130: The characteristics of the lamp used for irradiation are not clearly described in the Materials and Methods section. lines 131-144: The description of the ESR results (figure 2) is approximative and partially wrong: The authors affirm that two series of six hyperfine lines are present, but the parameters relative to the spectrum “b” are not defined. Therefore, the assignment to a DMPO-“sulfur” adduct cannot be acceptable. For a better spectral interpretation please see the paper from Zamora and Villamena (J. Phys Chem. 2012, doi: 10.1021/jp3039169) or similar references. Moreover, more information to the origin of spectrum “b” could be obtained analysing the similar and more intense spectrum “d”. I suggest to simulate all the spectra in order to make the correct assignments and to quantify the relative amount of each species. The statement “… the signal “b” was overlapped and weak, indicating lower reactivity between sulfur-centered radical and DMPO.” is not supported by any evidence. The different intensity of the two spectral pattern could be also due to a lower amount of S-radical or to a different affinity of DMPO for the two radical species. The authors should justify this affirmation or delete it. Figure S1 and S2 are not available. lines 157-159: The spectrum obtained after irradiation at wavelength > 520 nm is not only weaker, it is also different. Probably could be interesting to assign also this spectrum in order to better clarify the process. lines 214-215: On the basis of the previous remarks, I think that the sentence about the polymerisation mechanism is not justified by any experimental evidence. The authors should modify this affirmation or give more plausible explanations.   References: The paper concerning the spin-trapping technique are very old. Despite the pioneering works by Janzen and other authors are appreciated and still valid today, nevertheless they have been followed by 50 years of experiments with new insight that can help in dealing with these issues.

Author Response

Reviewer 1

The manuscript is well organized and the experimental design is appropriate. However, in my opinion, it should be improved before publication since the analysis of the experimental data is not accurate and the corresponding conclusion are not completely correct. 

In particular, the authors should address these specific remarks:

line 16 (and others in the text): Please substitute “noxide”  with “N-oxide”

Ans: Thank you for your comments. It has been revised.

lines 115-117: The experimental parameters are clearly wrong

Ans: Thank you for your comments. The parameters was set in equipment from a JEOL JES-X3 Series ESR spectrometer and it has been revised in Characterization section.

lines 129-130: The characteristics of the lamp used for irradiation are not clearly described in the Materials and Methods section.

Ans: Thank you for your comments. The characteristics of the lamp has been added in Characterization section.

lines 131-144: The description of the ESR results (figure 2) is approximative and partially wrong: The authors affirm that two series of six hyperfine lines are present, but the parameters relative to the spectrum “b” are not defined. Therefore, the assignment to a DMPO-“sulfur” adduct cannot be acceptable. For a better spectral interpretation please see the paper from Zamora and Villamena (J. Phys Chem. 2012, doi: 10.1021/jp3039169) or similar references. Moreover, more information to the origin of spectrum “b” could be obtained analysing the similar and more intense spectrum “d”. I suggest to simulate all the spectra in order to make the correct assignments and to quantify the relative amount of each species. The statement “… the signal “b” was overlapped and weak, indicating lower reactivity between sulfur-centered radical and DMPO.” is not supported by any evidence. The different intensity of the two spectral pattern could be also due to a lower amount of S-radical or to a different affinity of DMPO for the two radical species. The authors should justify this affirmation or delete it. Figure S1 and S2 are not available.

Ans: Thank you for your comments. We got the hyperfine coupling constants of the two series of six hyperfine lines by the simulation. We have deleted the statement “… the signal “b” was overlapped and weak, indicating lower reactivity between sulfur-centered radical and DMPO.” Figure S1 and S2 were showed in the supporting information.

lines 157-159: The spectrum obtained after irradiation at wavelength > 520 nm is not only weaker, it is also different. Probably could be interesting to assign also this spectrum in order to better clarify the process.

Ans: Thank you for your comments. The signal is a mess after enlargement and it is difficult to get valid information.

lines 214-215: On the basis of the previous remarks, I think that the sentence about the polymerisation mechanism is not justified by any experimental evidence. The authors should modify this affirmation or give more plausible explanations.  

Ans: Thank you for your comments. The adduct of DMPO and carbon-centered radical has been assigned according to literature, which supported the mechanism of the photolysis of the RAFT agent, though the sulphur-centered radical can not be confirmed owing to the lack of database of the same structure.

References: The paper concerning the spin-trapping technique are very old. Despite the pioneering works by Janzen and other authors are appreciated and still valid today, nevertheless they have been followed by 50 years of experiments with new insight that can help in dealing with these issues.

Thank you for your comments. We have added a reference recently reported using DMPO for spin trapping of a carbon-centered radical.

Reviewer 2 Report

Review of the manuscript entitled ‘ Investigation into the direct photolysis process of 2 photo-induced reversible addition fragmentation 3 chain transfer (RAFT) polymerization by electron 4 spin resonance (ESR) spin trapping’

The manuscript deals with reversible degenerative radical polymerization (RDRP) techniques. The authors use 5,5-Dimethyl-1-pyrroline noxide (DMPO) as a spin trap for the radicals generated from the direct photolysis of RAFT agent under visible light, aiming to prove the  direct photoactivation process. The motivation is that a spin trap by radical addition can form a more stable radical adducts easily detectable by ESR spectroscopy.

The manuscript is well written and the results well presented. A special attention must be done on the English spelling. A full revision must be done.

Author Response

Reviewer 2

Review of the manuscript entitled ‘ Investigation into the direct photolysis process of 2 photo-induced reversible addition fragmentation 3 chain transfer (RAFT) polymerization by electron 4 spin resonance (ESR) spin trapping’

The manuscript deals with reversible degenerative radical polymerization (RDRP) techniques. The authors use 5,5-Dimethyl-1-pyrroline noxide (DMPO) as a spin trap for the radicals generated from the direct photolysis of RAFT agent under visible light, aiming to prove the  direct photoactivation process. The motivation is that a spin trap by radical addition can form a more stable radical adducts easily detectable by ESR spectroscopy.

The manuscript is well written and the results well presented. A special attention must be done on the English spelling. A full revision must be done.

Thank you for your comments. The English has been improved now.

Round 2

Reviewer 1 Report

In the revised manuscript, the authors introduce some changes in the text in agreement with my previous comments. Despite most of the issues are now addressed, in my opinion a problem is still unresolved:  

lines 132-134: in the first revision of the manuscript I ask to the authors to modify the ESR parameters since they are clearly wrong. The authors thank me for the comment but they do not change anything, therefore the parameters remain wrong. Actually, it seems to me that the authors have a very limited experience in this technique if they do not realize that they have exchanged microwave frequency with modulation frequency, or if they indicate a scan range of 10 Gs and a modulation amplitude of 30 Gs for a spectra with a six-line pattern and hyperfine couplings of 13.90 Gs and 21.70 Gs. All the parameters must be corrected since they are inconsistent with experiments made in a X-band ESR spectrometer.

Minor points:

line 95: please correct “noxide” in "N-oxide" lines 220-222, caption of Fig 5: the presence of VCa in the experiments of Fig 5B is not reported, so the two series of spectra seems to be realized in the same experimental conditions.

Author Response

Response for Comments

In the revised manuscript, the authors introduce some changes in the text in agreement with my previous comments. Despite most of the issues are now addressed, in my opinion a problem is still unresolved:

lines 132-134: in the first revision of the manuscript I ask to the authors to modify the ESR parameters since they are clearly wrong. The authors thank me for the comment but they do not change anything, therefore the parameters remain wrong. Actually, it seems to me that the authors have a very limited experience in this technique if they do not realize that they have exchanged microwave frequency with modulation frequency, or if they indicate a scan range of 10 Gs and a modulation amplitude of 30 Gs for a spectra with a six-line pattern and hyperfine couplings of 13.90 Gs and 21.70 Gs. All the parameters must be corrected since they are inconsistent with experiments made in a X-band ESR spectrometer.

Response: Thanks for the comments. We are very sorry for the mistake of wrong parameter. The parameter was re-checked and revised as following in the revision: “General instrument parameters are as follows: microwave power, 1 mW, macrowave frequency, 9147 MHz, modulation frequency, 100 kHz, modulation amplitude, 300 G, field scan range, 100 G, center field, 3260 G.“

Minor points:

line 95: please correct “noxide” in "N-oxide" lines 220-222, caption of Fig 5: the presence of VCa in the experiments of Fig 5B is not reported, so the two series of spectra seems to be realized in the same experimental conditions.

Response: Thanks for these comments. We have revised the manuscript according to these comments.